# Basal State Calibration of a Chemical Reaction Network Model for Autophagy

**DOI:** 10.3390/ijms252011316

**Published:** 2024-10-21

**Authors:** Bence Hajdú, Orsolya Kapuy, Tibor Nagy

**Affiliations:** 1Department of Molecular Biology at the Institute of Biochemistry and Molecular Biology, Semmelweis University, 1085 Budapest, Hungary; hajdu.bence@phd.semmelweis.hu; 2Insititute of Materials and Environmental Chemistry, HUN-REN Research Centre for Natural Sciences, 1117 Budapest, Hungary

**Keywords:** biochemical reaction network, autophagy, parameter optimization, simulation

## Abstract

The modulation of autophagy plays a dual role in tumor cells, with the potential to both promote and suppress tumor proliferation. In order to gain a deeper understanding of the nature of autophagy, we have developed a chemical reaction kinetic model of autophagy and apoptosis based on the mass action kinetic models that have been previously described in the literature. It is regrettable that the authors did not provide all of the information necessary to reconstruct their model, which made their simulation results irreproducible. In this study, based on an extensive literature review, we have identified concentrations for each species in the stress-free, homeostatic state. These ranges were randomly sampled to generate sets of initial concentrations, from which the simulations were run. In every case, abnormal behavior was observed, with apoptosis and autophagy being activated, even in the absence of stress. Consequently, the model failed to reproduce even the basal conditions. Detailed examination of the model revealed erroneous reactions, which were corrected. The influential kinetic parameters of the corrected model were identified and optimized using the Optima++ code. The model is now capable of simulating homeostatic states, and provides a suitable basis for further model development to describe cell response to various stresses.

## 1. Introduction

Understanding the mechanisms of cellular death is crucial in the context of disease pathogenesis, as dysregulation of these processes is implicated in a wide range of conditions, including cancer, neurodegenerative disorders, and autoimmune diseases. For example, in cancer, impaired cell death pathways allow abnormal cells to survive and proliferate uncontrollably, while in neurodegenerative diseases, excessive cell death leads to the loss of vital neurons [1,2,3].

Cell death can be categorized into different types, based on the appearance of the dying cell. Two of these categories are apoptosis and autophagic cell death [4]. Apoptosis is an irreversible process, primarily responsible for eliminating abnormal or severely damaged cells. It also plays a key role in the removal of cells during embryonic development and the maturation of the immune system [4,5,6]. Apoptosis can be triggered by various forms of cellular stress, leading to the release of cytochrome c from the mitochondria, or by activation of death receptors. The former is known as the intrinsic pathway, while the latter is referred to as the extrinsic apoptotic pathway [7,8].

Autophagy, on the other hand, is present in cells at a basal level. During autophagy, damaged or long-lived proteins, misfolded proteins and abnormal organelles in the cytoplasm are engulfed in double-membrane vesicles called autophagosomes. These autophagosomes are then transported to lysosomes for degradation [9]. The degradation products, such as amino acids and sugars, are recovered and returned to the cytoplasm for reuse [10]. Autophagy is programmed to respond to various cellular stresses such as starvation, DNA damage, and hypoxia, helping cells to adapt to challenging conditions [11,12]. Although primarily a process that promotes cell survival, autophagy can also lead to cell death if the level of stress exceeds a critical threshold. In such cases, autophagy can promote cell death either by degrading essential cellular components or by facilitating the activation of apoptotic or necrotic programs [12]. Due to its complex regulatory nature, dysregulation of autophagy is associated with numerous diseases. In particular, in cancer, autophagy plays a dual role in both tumor promotion and tumor suppression, depending on the context [12,13].

Computational models of apoptosis have been used to predict how cancer cells respond to chemotherapy, and to identify potential mechanisms of drug resistance and strategies to overcome it [14,15,16,17,18,19]. Similarly, models of autophagy have been applied to understand its dual role in promoting both cell survival and cell death under different conditions, and to aid in the design of therapies that modulate autophagy for neurodegenerative diseases and cancer [20,21]. These models help to explore how subtle changes in cell death pathways can lead to different therapeutic outcomes, providing invaluable insights for clinical applications.

Building on these advances, more comprehensive models have been developed to capture the intricacies of cell fate decisions with greater precision. One such model by Liu et al. uses a chemical reaction network (CRN) framework to simulate cell fate decisions in apoptosis and autophagy [22]. A CRN defines a set of species and their reactions together with rate constants, allowing the mass action kinetic modeling of complex biochemical pathways through quantitative simulations [23]. However, the challenge is that rate constants of many reactions governing these pathways are not well known, requiring the use of data-driven techniques such as parameter estimation to approximate these values [24]. Even with parameter estimation, the quality of the model can be compromised by noisy data, often from heterogeneous cell populations, which can introduce bias and overlook cell-to-cell variability. Nevertheless, when both the structural properties of the network and the quality of the estimation are carefully considered, these models have the potential to predict the dynamic behavior of complex biological systems with significant accuracy [23,25].

### Features of the Initial Model

Liu’s model was selected because it is the most comprehensive and recent autophagy model we found in the literature. The model includes 94 species, five of which are designated as inputs, and inhibition or activation of different species was modeled by reducing their initial concentrations. The interactions between species are modeled using mass action kinetics, resulting in a reaction network with 129 highly uncertain rate constants. The model can be divided into five smaller submodules: Calcium, Inositol, mammalian target of rapamycin (mTOR), Autophagy, and Apoptosis.

The apoptosis module is based on a previous publication of Liu, where they investigated the nuclear p53 transcriptional activity under genotoxic stresses [26]. Under genotoxic stress, nuclear p53 activates transcription of pro-apoptotic proteins BAX and p53 upregulated modulator of apoptosis (PUMA), while inducing Mouse double minute 2 (Mdm2), which inhibits p53 by promoting its ubiquitination and translocation to the mitochondria. Mitochondrial p53 inhibits Bcl-2 and activates BAX, leading to cytochrome c release and caspase activation, amplified by a feedback loop involving truncated BH3 interacting-domain death agonist (tBid), with Endoplasmic Reticulum (ER) stress also triggering apoptosis through c-Jun N-terminal kinase (JNK)-, protein kinase R (PKR)-like endoplasmic reticulum kinase (PERK) -, Death-associated protein kinase 1 (DAPK1)-, and ${\mathrm{Ca}}^{2+}$-dependent pathways [22,26].

Autophagy is regulated by 5’ AMP-activated protein kinase (AMPK) and mTOR coplex 1 (mTORC1) through Unc-51-like autophagy-activating kinases 1 (ULK1). Under nutrient-rich conditions, mTORC1 represses autophagy by phosphorylating ULK1, whereas AMPK promotes autophagy during nutrient limitation by directly phosphorylating ULK1 and inhibiting mTOR via the Tuberous sclerosis proteins 1 and 2 (TSC1/2) pathway or by directly phosphorylating the subunits of the mTORC1 complex [27,28]. In addition, ULK1 has feedback loops with both mTOR and AMPK. It represses mTOR activity, and also downregulates AMPK activity [27,29]. In addition, autophagy and apoptosis are closely linked through interactions with UV radiation resistance-associated gene protein (UVRAG), Bcl-2, caspases, p53, ER stress, and calpain, which regulate both processes in response to cellular stress [22,30,31].

Calpain activity depends on the intracellular ${\mathrm{Ca}}^{2+}$ concentration [32]. The ${\mathrm{Ca}}^{2+}$ potential between the ER and the cytoplasm is regulated by voltage-gated and ligand-gated ion channels, such as Inositol trisphosphate receptor (IP3R), and pumps, such as sarcoplasmic/endoplasmic reticulum ${\mathrm{Ca}}^{2+}$-ATPase (SERCA) [22,33]. High cytoplasmic concentrations of ${\mathrm{Ca}}^{2+}$ activate Calcium/calmodulin-dependent protein kinase kinase 2 (CaMKKβ), which activates AMPK and indirectly induces atuophagy. CaMKKβ also activates calpain [22,34]. Calpain activity leads to autphagy protein 5 (ATG5) cleavage via the inositol pathway, resulting in autophagy suppression, and it can also induce apoptosis via activation of BAX [35,36].

The intracellular ${\mathrm{Ca}}^{2+}$ concentration also depends on the cyclic adenosine monophosphate (cAMP) levels in the cells [37]. The level of cAMP is regulated by the G protein-coupled receptor (GPCR) signaling pathway [38,39]. Elevated levels of cAMP lead to the activation of phospholipase Cϵ (PLCϵ), which subsequently produces increased levels of inositol trisphosphate (IP3), which is a cofactor of IP3R, facilitating the release of ${\mathrm{Ca}}^{2+}$ ions from the ER [38,39].

The model’s parameters were calibrated with image-based single-cell experimental data obtained from Xu et al.’s study [40]. The cells were exposed to various stress-inducing conditions, including staurosporine (STS), rapamycin, and tunicamycin, to observe the differential dynamics of these cellular processes. The data series measured only two outputs: cell death levels and autophagy levels [40]. Although the estimated parameter values were reported in the article, the initial concentrations of the species and the sampling intervals used for the simulations were not published.

In order to build upon the model proposed by Liu et al., we contacted the authors of the article in question [22]. However, we were unable to obtain the model or the initial concentrations used for the simulations, as the authors were no longer in possession of this information. While the model reactions and corresponding chemical reaction constants are provided in the Appendix A, the initial concentrations and their sampling ranges are absent, rendering the model irreproducible. Furthermore, in the original study by Liu, the behavior of the model in an unperturbed, basal state was not discussed. It is essential to know a dynamical system’s unperturbed state in order to validate the model. In this study, the model was implemented in multiple modeling frameworks and plausible initial concentration ranges for each species were defined. Subsequently, the model parameters were reevaluated, enabling the model to simulate a homeostatic state.

## 2. Results

### 2.1. Initial Protein Concentrations

To be able to run simulations on the model, we had to search the literature for new concentration values. We found several models that were published with initial concentrations, each of which is associated with at least one submodule of the base model, as detailed in Table 1. These models are quantitative in nature, with each differential equation representing the temporal evolution of a specific state variable, namely the concentration of a biomolecule. Furthermore, these models include initial concentrations for each species, which we used to compile the data presented in Appendix A. We also took into account the different cell types and assumed a cell volume of $10^{-12}$ L. A cell volume had to be defined, since the protein quantities in some sources were expressed as counts per cell rather than in concentrations. For the stationary state, we assumed a low level of autophagy, as described by [41], with apoptosis and its inducers completely inactive. We also assumed that the cell is proliferating, which implies that mTOR is active [42,43]. The intracellular calcium ion concentration was set in the 100 nM (i.e., $100\times 10^{-9}\mathrm{mol}/{\mathrm{dm}}^{3}$) range, while the calcium ion concentration in the endoplasmic reticulum (ER) was assumed to be in the μM range [38,39]. In addition, the concentrations of all complexes were assumed to be zero at t=0. The initial concentration ranges for all species and their corresponding sources are provided in Appendix A.

The sources presented in Table 1 used several different strategies to define initial concentrations. Dalle et al. define the concentration of several species as 0, thereby simplifying the model optimization process [44]. Bagci et al.’s approach was similar to Dalle’s; they only defined nonzero value to a selected few species [19]. Sundaramurthy et al. assumed that the concentration of kinases that regulate multiple signaling pathways is greater than that of proteins with more limited specificity [45]. If the source used artificial units, like Tavassoly et al., we were still able to use the data, but with careful adjustments [46]. Instead of relying on the absolute values, we focused on the relative proportions of the species. This allowed us to preserve the balance between different proteins in the system, even though the exact concentrations were not directly applicable.

**Table 1 ijms-25-11316-t001:** Models used for the definition of the initial protein concentration ranges. R stands for the number of reactions in the given model.

Mechanism	R	Cell Type	Relevant Species	Source
Apoptosis	23	HeLa	procaspase	[15]
Apoptosis	25	HeLa	procaspase, BID, tBID	[16]
Apoptosis	24	E.coli	cyt c, BIT, Bax, procasp	[19]
Apoptosis	20	HeLa	p53, MDM2	[17]
Apoptosis	19	MCF7	PUMA, Bax, BCL2 and their complexes	[18]
Apoptosis	5	MCF7	p53, MDM2	[14]
${\mathrm{Ca}}^{2+}$ signaling /Ras	43	not specified	Ca ions, SERCA, PIP_2, PLC, IP_3	[38]
${\mathrm{Ca}}^{2+}$ signaling/Ras	11	eukaryotic	Ca ions, G-proteins, PLC, IP_3	[39]
JNK/p38 cas	12	not specified	JNK, MAPK	[45]
Autophagy and apoptosis	13	RTP	ATG5, autophagosomes, BCL2, BEC1, Ca ions, CALPAIN, caspase, DAPK, PI3R, JNK, mTOR	[46]
mTOR signaling	25	HeLa	AKT, mTOR, TSC1/2, PI3K	[44]
mTOR signaling	18	HeLa	AKT, mTOR	[47]
EGFR signaling	129	NSCLC	PIP2, AKT	[48]
Ras signaling	6	SB2 melanoma	PKC, cAMP, G-proteins, PIP2, PKA, MAPK,	[49]
mTOR signaling	13	Human	mTOR	[50]

To achieve plausible initial value ranges for each species, we combined all the methods and simplifications introduced by the models shown in Table 1. If we have found no data for a given species, we set its range between 0 and 100 nM, thus taking into account the possible uncertainty of the variables. Furthermore, it was considered that the number of kinases may be greater than that of proteins involved in the execution of apoptosis. In the event of multiple initial values for a given protein, the sampling range was defined in a manner that ensured all values were contained within the specified range. The resulting concentration ranges are shown in Appendix A.

### 2.2. Revision of Incorrect Reactions

In order to achieve a sensible simulation, the reactions of the model were modified, as we found several incorrect reactions in the model. In the original model, ER stress activated both PERK and JNK. Upon activation, PERK triggered activating transcription factor 4 (ATF4), which subsequently induced phagophore formation. In contrast, JNK phosphorylates Bcl-2, leading to a dual effect: phosphorylated Bcl-2 is unable to form complexes with Beclin-1 (BECN1), thereby promoting autophagy, but it also releases BAX, which can promote apoptosis [22]. However, the interactions of these proteins with autophagy and apoptosis are more complex than captured in the model.

During the unfolded protein response (UPR), PERK inhibits protein synthesis by phosphorylating eukaryotic translational initiation factor 2α (eIF2α), thereby reducing the load on the endoplasmic reticulum [51]. This inhibition leads to the activation of the transcription factor ATF4, which upregulates several autophagy-related genes (ATGs), including microtubule-associated proteins 1A/1B light chain 3B (LC3B), ATG5, ULK1, and BECN1 [52]. Under tunicamycin-induced ER stress, LC3 transcription is stimulated by ATF4, suggesting that PERK can indeed promote autophagy through ATF4 activation. However, PERK also plays a dual role in mediating apoptosis during prolonged ER stress, as ATF4 regulates the expression of C/EBP homologous protein (CHOP), a key pro-apoptotic factor [53,54,55,56,57]. Evidence suggests that downstream of mTOR, PERK activation can inhibit autophagosome–lysosome fusion in response to chronic ER stress, thereby contributing to ER stress-induced apoptosis and further complicating the interactions of PERK with other species in the model [54]. In addition, several studies have shown that JNK can also have multiple pro-autophagy effects under certain ER stress conditions [58,59,60,61]. Expanding these interactions would significantly increase the number of parameters, thereby reducing the discriminative power of the model.

The roles of protein kinase A (PKA) and protein kinase C (PKC) have also been modified; in the published model, PKA helps to form phagophores, which is the opposite of what it actually does. It is true that PKA can phosphorylate LC3, but this phosphorylation prevents the recruitment of LC3 to phagophores, thereby inhibiting autophagy [62]. In the original model, PKA was activated during nutrient stress conditions through mitogen-activated protein kinase 15 (MAPK15), with no connection to the cAMP signaling pathway. However, as PKA activity is closely dependent on cAMP, we introduced a response that activates PKA via cAMP [63,64,65]. As the only function of MAPK15 in the model was to activate PKA, we decided to exclude MAPK15 from the model to streamline the system. Similarly to PKA, PKC also had an opposite effect on autophagy in the model, helping to form phagophores.

The levels of cAMP in the model depended solely on the concentration of GPCRs acting as inputs. Consequently, there was no feedback loop to regulate cAMP levels. Based on the findings of Dolan et al., that PKC controls phospholipase C activity, we incorporated this feedback mechanism into the model [66].

The reactions of mTOR were also modified. While protein kinase B (PKB/AKT) is generally expected to have a positive effect on the activity of the mTORC1 complex, the reaction mTOR: ULK + AKTA → mTOR + ULKA + AKTA suggests that AKT disrupts the mTORC1 complex, resulting in an active ULK1 kinase and an inactive mTORC1 complex [67]. While it is true that mTORC1 can transiently complex with ULK1 during phosphorylation, we have revised the reaction to reflect that this interaction is transient and occurs as part of a chemical reaction, rather than simple association/dissociation dynamics [28,68,69]. Following the implementation of the aforementioned modifications, the final revised model comprises 113 reactions and 84 species.

### 2.3. Simulating the Basal State

In order to evaluate the initial model’s ability to predict the basal state of cells, 20 sets of initial concentrations ($N_{\mathrm{cond}}=20$ conditions) were generated by uniform random sampling of concentrations within the defined feasible concentration ranges in Appendix A. These scenarios were simulated for one day (i.e., 24 h in cell life) within the Optima++ simulation environment [70,71,72] using the Cantera code [73], which is a popular solver in chemical reaction kinetics. As a result of the simulations, concentration profiles were obtained for all the 84 species of the mechanisms.

The state of the system was characterized by the concentration of 34 selected species: Adenylyl cyclase (AC), active form of AKT (AKTA), AMPK, ATG5, truncated ATG5 (ATG5T), Bcl-2 (BCL2), BCL2 and BAX complex (BCL2_BAX), BCL2 and PUMA complex (BCL2_PUMA), BEC1, BAX, BH3 interacting-domain death agonist (BID), the Ca ion concentration in the ER (CA2ER), and in the cytoplasm (CA2IC), CAMKKβ, CALPAIN, DAPK, exchange protein activated by cAMP (EPAC), active from of GPCRs (GPCRA), G protein subunit α (GA), G protein subunit βγ (GBC), IP3, Phosphatidylinositol 4,5-bisphosphate (PIP2), PKA, PKC, Inactive Phospholipase C epsilon 1 (PLCE), Active Ras Homolog Enriched In Brain (RHEBA), SERCA, Inactive TSC1/2 (TSC), ULK1 (ULK), UVRAG (UVG), cytochrome c in the mitochondria (CYTCM), active mTOR (MTORA), P53, and procaspase (PROCASP). Information on the concentrations of these species was available from the models listed in Table 1. In the basal state of the cells, the concentrations of these species stay within these ranges; therefore, the ability of a model to keep the corresponding concentrations within these ranges can be used to assess its performance.

To measure the deviation of the model solution from the center of the basal state, the following error function (*E*) was proposed, which measures the root-mean-square deviation (RMSD) of the simulated concentrations ($c_{s,i}^{\mathrm{sim}}\left(t_{j}\right)$) from the mean concentrations of these species (for species *s*: $c_{s}^{\mathrm{mean}}$):(1)$$E=\sqrt{\frac{1}{N_{\mathrm{spec}}\cdot N_{\mathrm{cond}}\cdot N_{\mathrm{tim}}}\sum_{s=1}^{N_{\mathrm{spec}}}\sum_{i=1}^{N_{\mathrm{cond}}}\sum_{j=1}^{N_{\mathrm{tim}}}\frac{{\left(c_{s,i}^{\mathrm{sim}}\left(t_{j}\right)-c_{s}^{\mathrm{mean}}\right)}^{2}}{\sigma_{s}^{2}}}$$

Here, $N_{\mathrm{spec}}=34$ is the number of species, $N_{\mathrm{cond}}=20$ is the number of conditions (i.e., the different sets of initial concentrations), $N_{\mathrm{tim}}=25$ is the number of time points (i.e., $t_{j}$ at every hour in a day), whereas s,i,j are the corresponding running indices for the sums. Standard deviation $\sigma_{s}$ denotes one eighth of the width of concentration range (i.e., $\left[c_{s}^{\mathrm{mean}}-4\sigma_{s};c_{s}^{\mathrm{mean}}+4\sigma_{s}\right]$) for species *s*, which can be calculated as the difference of the maximum and the minimum values of the feasible concentration. Consequently, this error formula penalizes those simulation results which deviate significantly (i.e., with multiple $\sigma_{s}$ from the mean value, which is defined as the average of the minimum and maximum concentrations). To assess the model performance in detail, the following species-specific error function is defined and calculated for each individual species (*s*),
(2)$$E_{s}=\sqrt{\frac{1}{N_{\mathrm{cond}}\cdot N_{\mathrm{tim}}}\sum_{i=1}^{N_{\mathrm{cond}}}\sum_{j=1}^{N_{\mathrm{tim}}}\frac{{\left(c_{s,i}^{\mathrm{sim}}\left(t_{j}\right)-c_{s}^{\mathrm{mean}}\right)}^{2}}{\sigma_{s}^{2}}}$$

This implies that the total error can be calculated from species errors,
(3)$$E=\sqrt{\frac{1}{N_{\mathrm{spec}}}\sum_{s=1}^{N_{\mathrm{spec}}}E_{s}^{2}}$$

If there were no reactions, the initial distribution of concentrations would stay constant, and the $N_{\mathrm{cond}}\to \infty$ limit would give a $\sqrt{\int_{-4}^{+4}x^{2}/8}=4/\sqrt{3}\approx 2.3$ error for each $E_{s}$, and also for *E*. The calculated $E_{s}$ error for the species over the 20 simulations are shown in Table 2, which shows that the initial model overestimates BAX concentrations by several orders of magnitude ($E_{s}∼10^{10}$, and significant errors were observed also for more than half of the species ($E_{s}∼3-40$), whereas the rest of the species were around or below the statistical constant value (∼2.3). This suggests that the initial model contains several rate parameters that need to be adjusted significantly in order for the model to keep concentrations within the basal range.

### 2.4. Identification of Influential Rate Coefficients

In order to identify which model parameters need to be calibrated, we carried out a local sensitivity analysis by perturbing the rate coefficient ($k_{n}$) of each of the $N_{\mathrm{reac}}=$113 reactions (14 reversible pairs (2 × 14) and 85 one-way reactions) by +5% (i.e., 1.05$\cdot k_{n}$), and simulated all 113 modified models for all 20 conditions. The concentration-time solution of these perturbed simulations are denoted as $c_{s,i}^{\mathrm{sim}}\left(t_{j};1.05\cdot k_{n}\right)$. By making finite differences, σ-normalized local sensitivity coefficients $S_{s,i,n}$ were calculated as finite differences:$$S_{s,i,n}\left(t_{j}\right)=\frac{1}{\sigma_{s}}\frac{\partial c_{s,i}^{\mathrm{sim}}\left(t_{j}\right)}{\partial \ln k_{n}}\approx \frac{1}{\sigma_{s}}\frac{c_{s,i}^{\mathrm{sim}}\left(t_{j};1.05\cdot k_{n}\right)-c_{s,i}^{\mathrm{sim}}\left(t_{j}\right)}{\ln1.05}$$

A reaction has significant effect on a species concentration (upon parameter perturbation) if the absolute value of this dimensionless sensitivity is significant, i.e., larger than 0.01. An overall impact ($I_{n\to s}$) of the perturbation of $k_{n}$ on the concentration profiles of the $s^{th}$ species can be obtained by taking the root mean square averages of these local sensitivities and multiplying them with standard deviation of the parameter uncertainty ($s_{n}$) for $\ln k_{n}$,
(4)$$I_{n\to s}^{\mathrm{SUE}}=\sigma_{n}\cdot \sqrt{\frac{1}{N_{\mathrm{cond}}\cdot N_{\mathrm{tim}}}\sum_{i=1}^{N_{\mathrm{cond}}}\sum_{j=1}^{N_{\mathrm{tim}}}S_{s,i,n}^{2}\left(t_{j}\right)}$$

This impact measure was proposed by Kovács et al. ([74]), and its advantageous feature is that, beyond local sensitivities (“S” superscript), it also takes into account the uncertainty of the parameters (“U” superscript) and experimental errors (“E” superscript). In this study, as no information is provided on the uncertainty of the model parameters, the same large, ±4 orders of magnitude uncertainty was assumed for all reactions. This uncertainty corresponds to $f_{n}=4$ uncertainty parameter value, which is defined as the half-width (radius) of the uncertainty range on 10-based logarithmic scale (see, e.g., [75]):$$f_{n}=\log_{10}\frac{k_{n}^{\max}}{k_{n}^{0}}=\log_{10}\frac{k_{n}^{0}}{k_{n}^{\min}}$$
where $k_{n}^{0},k_{n}^{\max}$ and $k_{N}^{\min}$ are nominal, the maximum and the minimum physically possible (i.e., estimated range based on our present knowledge on the reaction) values for $k_{n}$. In this study, the same uncertainty was assumed for all reactions, thus it did not affect the ranking of reactions based on the $I_{n\to s}^{\mathrm{SUE}}$ values. The reaction ranking found the following two reactions as the most important:$$\begin{array}{cc} P53A & \overset{k6}{\to }P53A+\mathrm{BAX}\\ \mathrm{BAX} & \overset{k12}{\to }⌀ \end{array}$$
which is probably due to their being directly responsible for the large overprediction of BAX concentrations. Furthermore, the analysis found that 99 additional reactions have significant impact values. Finally, the remaining 12 reactions are active in various stress events, thus were dormant in the basal state and, accordingly, showed zero impacts on the concentration profiles of the 34 species. Consequently, the 34 concentration profiles in the 20 simulations from random initial conditions are determined by all other reactions, thus it seems they provide a wealth of information to constrain the model. The ranking of reactions with their $I_{n\to s}^{\mathrm{SUE}}$ values are available in the Appendix A.

### 2.5. Optimization of Influential Rate Coefficients

Using code Optima++ with the FOCTOPUS numerical optimization algorithm [70,71,72] and the CANTERA solver, we minimized the *E* overall error function by tuning the rate coefficients of the 101 influential reactions within ±4 orders of magnitude uncertainty range. The optimization could reduce the overall *E* error greatly from $1.3\times 10^{9}$ to 3.16; however, $E_{s}$ errors could not be reduced for all species, and multiple species had errors above 7. Examining the optimized rate coefficients, it was noticed that several had values optimized to the edge of the uncertainty range, and had become ineffective, as they either switched off the reaction (i.e., too low $k_{n}$), or consumed its reactants too quickly (i.e., too high $k_{n}$). To bring these parameters back to the sensitive range, their parameters were set to the initial value, and the optimization was restarted. The optimization was performed altogether in four passes, and the evolution of the *E* error function in each pass as a function of number of iterations can be seen in Figure 1. After the third pass, only four species (BAX, IP3, PIP2, and PKC) had significant errors, and it turned out that for the rate coefficients of reactions that consumed species IP3 and PKC, some were optimized to too fast values (e.g., IP3, PKC were consumed within 1 h), thus they were set back to their starting value manually. These modifications immediately gave a smaller error (2.93 vs. 2.74), and with a subsequent optimization, the error could be reduced down to 2.22. Additional manual adjustments and the fifth optimization pass could not reduce the overall error any further.

The errors for the individual species were also calculated, and together with their initial values, can be seen in Table 2. For all species that had significant errors (i.e., $3-2.1\times 10^{10}$) in the initial mechanism, the optimization greatly improved their description as their errors dropped down below 2.60, except for BAX ($E_{S}=3.37$), whose initial error was out of charts. Furthermore, the species with good initial description maintained their low errors.

BAX plays a central role in apoptosis, which does not occur under stress-free conditions, and in homeostasis, which is the scope of the present investigation. Accordingly, the initial BAX concentrations were sampled to be very low (~30 nM) in order to test the dynamics of the model close to the basal state. Although the model does not give the best possible fit for BAX, it does keep BAX concentrations at low levels in line with the physiological behavior.

The 20 concentration vs. time curves ($c_{s,i}^{\mathrm{sim}}\left(t_{j}\right)$, j=0⋯24) for four key species—active mTOR (MTORA), inactive ULK (ULK), procaspase (PROCASP), and ${\mathrm{Ca}}_{\mathrm{ER}}^{2+}$ (CA2ER)—were selected due to their critical roles. Active mTOR indicates cell growth, while inactive ULK signals autophagy activity. A constant procaspase concentration implies no apoptosis, and depletion of ER ${\mathrm{Ca}}^{2+}$ can trigger both autophagy and apoptosis, making its concentration a crucial factor in the model. Simulations comparing the initial and optimized mechanisms are shown in Figure 2. For the rest of the species (i.e., 30), they can be found in the Appendix A. The curves show that while the initial model leads to inactive mTOR, active ULK states and procaspase cleavage, the optimized mechanism sustains their concentrations within the physiological range. Concentration profiles for all 34 species are available in the Appendix A.

The optimization not only provided a better model, but also allowed us to greatly constrain the uncertainty of the optimized parameters using the formula for estimating the posterior covariance matrix of parameters by Turányi et al. [70]. Table 3 contains the initial and optimized values of the rate parameters in ten-based logarithmic scale and the posterior uncertainty parameters ($f^{\mathrm{posterior}}$) and uncertainty factors ($10^{f^{\mathrm{posterior}}}$) for 19 reactions whose rate coefficient could be constrained within one order of magnitude. The changes in parameter values and the uncertainties can be found for all reactions in Appendix A, which shows that the basal concentration range can constrain the rate coefficients of 19, 72, 6, and 1 reactions in 0–1, 1–2, 2–3, and 4–5 orders of magnitude uncertainty, respectively. These ranges can serve as prior uncertainties for latter model developments.

The defined constraints used for parameter optimization are based on basal protein levels in the system. As a result, proteins involved in apoptosis and autophagy are represented at much lower levels, and reactions involving these proteins cause minimal perturbations in the model output, making their parameter optimization less certain, as shown in Table 3. For example, although ATG5 is involved in autophagy, it has limited activity in homeostatic states. Specifically, it promotes autophagy by binding to BCL2 in its truncated form, making its role in this optimization problem well defined and allowing more precise tuning of the parameters. Furthermore, the background activation and deactivation of other proteins could also be optimized with greater precision, indicating that the basal conditions of the system rely on mechanisms that are not fully represented in the model. This suggests that the effects of these reactions can only be optimized if these mechanisms were defined in the model more precisely.

## 3. Discussion

The omics technologies generate comprehensive molecular profiles of cells across a range of biological scales, resulting in intricate interaction maps and extensive databases [76]. The resulting data are frequently analyzed using statistical methods to identify patterns within these large datasets, which often involve complex, nonlinear associations between data subgroups [20,77]. This results in black-box models that offer little insight into the mechanistic relationships between input features and predicted outcomes. This lack of interpretability limits their broader applicability, making it challenging to derive biologically meaningful insights and validate predictions experimentally [77].

To address these limitations, the use of dynamical models is essential. One such model is Liu’s model, which comprises five modules representing key signaling cascades activated by nutritional, genotoxic, and endoplasmic reticulum (ER) stresses. It contains 94 components that represent different states of protein activation, binding or localization. The system dynamics are governed by a set of ordinary differential equations (ODEs) integrating mTOR, inositol signaling, autophagic, and intrinsic apoptotic pathways. Crosstalk between these pathways is mediated by Bcl2, caspases, p53, and calpain, and modulates autophagic and apoptotic responses through feedback loops involving G protein signaling and CaMKKβ [22]. In addition, AMP-activated protein kinase (AMPK) is shown to fine-tune autophagic and apoptotic responses through feedback loops involving G-protein signaling and CaMKKβ [22]. Unfortunately, the authors have not released the source code for their model, and were unable to provide it upon request. Consequently, we had to define the initial concentrations ourselves, as they were not published.

The model is divided into five sub-modules, several of which correspond to existing models in the literature, as detailed in Table 1. Using these references and other literature sources, we compiled a plausible set of initial concentrations for each species, which are provided in the Appendix A. Using these values, we reconstructed the model, which is now available on GitHub (https://github.com/mcsksgyrk/basal_state_calibi accessed on 13 October 2024).

During simulations of the reconstructed model, we observed that it failed to maintain a basal state, even in the absence of stress, as both apoptosis and autophagy were activated regardless of the initial conditions. In dynamical systems theory, it is equally important that the system under investigation is capable of maintaining an unperturbed stationary state. In biological dynamical systems, this is known as homeostasis, where the system maintains a stable state that corresponds to a healthy normal state [77]. Further investigation revealed several incorrect equations, including errors in the roles of PKA and PKC, which incorrectly promoted autophagy instead of inhibiting it, and AKT, which had an unexpected negative effect on mTOR activity. PERK and JNK proteins were excluded from the revised model to simplify the study’s focus on finding the basal state. Despite these corrections and simplifications, the model still failed to maintain a basal state. We believe this may be due to the fact that the model was developed using only a single data source and tested only under specific stress conditions. With over 100 unknown parameters and limited training data, the model is likely to be overfitted.

To quantify the deviation from the basal state, we measured the root mean square deviation (RMSD) of the simulated concentrations from the mean concentrations of key species. The simulations showed that the model significantly overestimated BAX concentrations by several orders of magnitude, along with other species that exhibited notable errors. Local sensitivity analysis revealed that in total, 101 reactions have a significant impact on the baseline conditions of the system, indicating that several rate parameters in the original model may require adjustment. Consequently, a new parameter estimation was performed using the FOCTOPUS algorithm and the CANTERA solver within the Optima++ environment. The new optimization significantly reduced the overall error from $1.3\times 10^{9}$ to 2.2, but this required some manual intervention as the rate coefficients were temporatily optimized to the edge of their uncertainty range.

We successfully defined the initial concentrations and calibrated the model to achieve a stable, unperturbed steady state. While quantitative models are invaluable tools for understanding biochemical systems, there are significant challenges in developing robust and reliable models. Sensitivity analysis and careful parameter tuning are essential to ensure stability and accurate biological representation. This model advances our understanding of the interplay between autophagy and apoptosis under ER stress, a condition implicated in many common diseases such as neurodegenerative disorders and diabetes. In the future, we will build on the results of the present work; the scope of our next studies will be the modeling of cell behavior under different stresses, where BAX is expected to exhibit much more vivid dynamics, the description of which will be much more critical, and will provide stronger constraints for setting the rate parameters of its reactions.

One of the greatest challenges in modern medicine is delivering effective personalized therapies that are precisely tailored to an individual’s unique biology or biological state [78]. The advent of omics technologies has greatly advanced the field of personalized medicine by providing detailed molecular profiles. However, while these methods help identify causal relationships, they alone are insufficient for predicting the effects of therapies at an individual level. To achieve truly accurate predictions, the development of mechanistic models is crucial for processing and interpreting the vast amounts of data generated by omics technologies [77,79]. Dynamical models are particularly beneficial in this context, as they allow for the prediction of the system’s stationary states (fixed points) and help determine the possible trajectories or states the system may evolve towards [25,77]. The advanced model has made the original framework reproducible and readily available for future use, enabling other researchers to easily expand its reactions and adapt it for simulating various treatments. By providing a robust and flexible tool, we facilitate more refined simulations in the field of personalized medicine, which we hope will contribute to a deeper understanding of the intricate mechanisms of autophagy regulation.

## 4. Materials and Methods

### Mathematical Modeling

The interactions between species in the model were described using mass action kinetics, resulting in a system of ordinary differential equations (ODEs). The numerical calculations were performed using Optima++, a framework [70,71] designed for model validation, sensitivity analysis, and parameter optimization in large reaction kinetic problems. The ODEs were solved using the Cantera solver [73], which efficiently handles even large chemical kinetics systems, allowing us to swiftly simulate the temporal evolution of even hundreds of species concentrations under different conditions.

The Optima++ code can minimize a root-mean-square type error function, which measures the deviation of simulation results from experimental data or from other reference values. The Optima++ code was original developed for temperature-dependent problems, and accordingly it can tune parameters of the Arrhenius equation, $k\left(T\right)=AT^{n}\exp\left(-E/RT\right)$, which can describe the temperature dependence of the rate coefficient in a wide temperature range. However, most living organisms thrive in a narrow temperature range, thus biology models are usually temperature-independent and the rate coefficient of each reaction is characterized by only a single number. Accordingly, this single parameter value was assigned to the pre-exponential factor *A*, while the other two parameters, *n* and *E*, were set to zero. The optimization is carried out on logarithmic scale (i.e., lnA) to efficiently cover all orders of magnitude during parameter sampling.

For the optimizations, we used Optima++’s FOCTOPUS algorithm, which was benchmarked in temperature-dependent combustion kinetic problems and found to be more robust than any of the popular global optimization algorithms [72].

The FOCTOPUS name is an acronym, which stands for FOCusing robusT Optimization with Uncertainty-based Sampling. The method employs either random uniform or Gaussian sampling of parameters within the user-defined prior uncertainty ranges. In the first iteration, *N* samples taken in the whole prior uncertainty range, and if a lower objective function value (e.g., model fitting error) is found, the algorithm centers the sampling distribution around it, and takes another set of *N* samples. If no improvement is found, then its divides the sampling volume (i.e., in the multidimensional parameter space) by *N*, thereby focuses on the actual best solution, and takes another set of *N* samples. If improvement is found, it sets the center of the sampling distribution to this point and zooms out by increasing the sampling volume by a factor of *N*, unless the original volume of the multidimensional prior uncertainty range is reached. It keeps repeating the procedure until all parameters gets sufficiently converged, which requires that the sampling radius in each parameters drops to a very small fraction of the prior uncertainties.

Selection of active parameters are usually based on local sensitivity analysis. In this study, we used a more advanced method, called SUE [74], which also considers reference data and parameter uncertainty information beside local sensitivities, and takes into account the actual form of the error function. Formulae tailored for this study for parameter selection and for the error function, and the definition of prior uncertainty ranges are presented in the result chapter.

The used models were implemented in julia, which is available on a code hosting platform GitHub (https://github.com/mcsksgyrk/basal_state_calibi, accessed on 10 September 2024). 

## Figures and Tables

**Figure 1 ijms-25-11316-f001:**
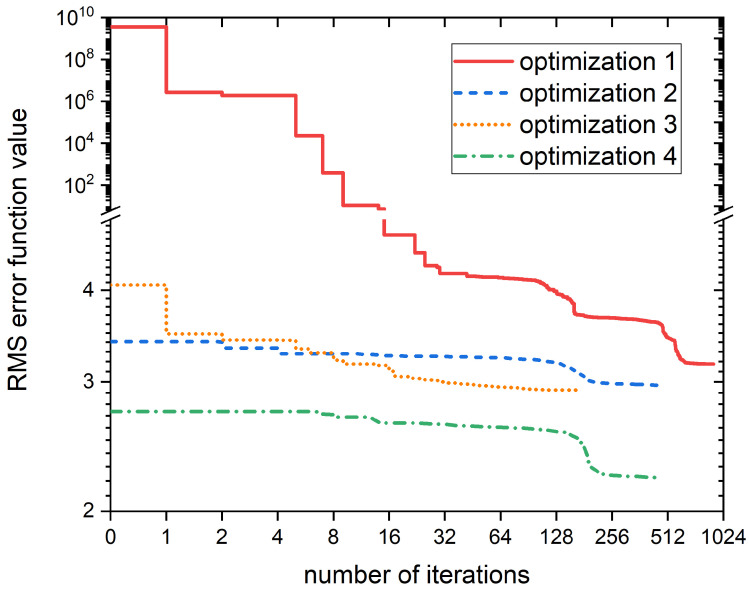
Evolution of the overall *E* root-mean-square concentration error (for 34 species in one-day simulations from 20 conditions) as a function of number of iterations in the four optimization passes. There were manual adjustments after passes 1–3 to move certain rate coefficients into the effective range, thus the subsequent passes started from different error values. Note that axes are given with logarithmic scale, and there is a scale break in the y-axis at value 5.

**Figure 2 ijms-25-11316-f002:**
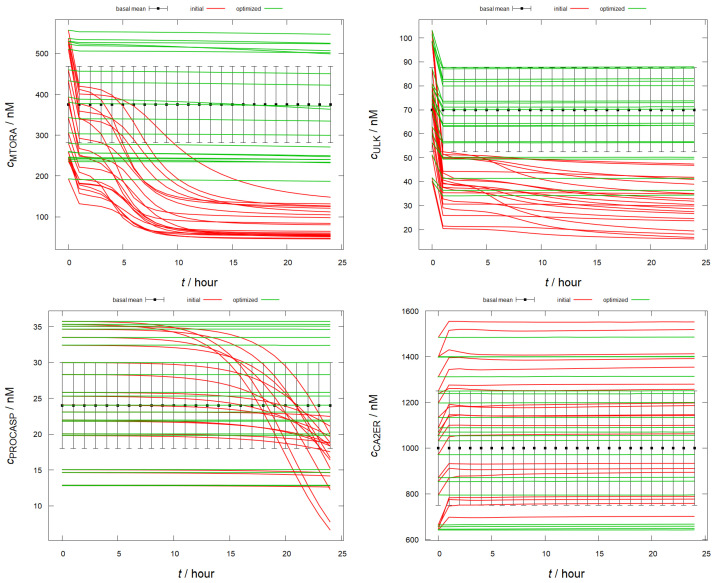
Concentration profiles from 20 random initial conditions within the $\left[c_{s}^{\mathrm{mean}}-4\sigma_{s};c_{s}^{\mathrm{mean}}+4\sigma_{s}\right]$ basal range for selected species simulated with the initial (red) and the optimized (green) mechanisms over 24 h. The center of the basal range with $2\sigma_{s}$ error bars (e.g., $c_{s}^{\mathrm{mean}}\pm 2\sigma_{s}$) are also shown.

**Table 2 ijms-25-11316-t002:** Comparison of species-specific root-mean-square errors (one-day simulations of 20 conditions) of the concentration-time profiles simulated with the initial and final optimized mechanisms, sorted from highest (red) to lowest (green). Species are referred to by their name in the model.

Species	ini	opt	Species	ini	opt
BAX	2.09 × $10^{10}$	3.37	CAMKKB	2.72	2.24
BCL2_BAX	39.0	2.24	DAPK	2.63	2.63
UVG	7.85	2.39	PROCASP	2.38	2.56
BCL2	7.85	1.81	PIP2	2.46	2.32
BCL2_PUMA	7.83	2.26	AC	2.35	2.36
AKTA	7.71	2.20	CALPAIN	2.36	2.23
BEC1	7.22	2.27	GPCRA	2.34	2.34
ATG5T	7.01	1.67	GA	1.99	2.28
RHEBA	6.63	2.08	PKA	2.25	2.26
CA2IC	6.40	2.09	P53	2.24	2.24
ATG5	5.93	1.14	CA2ER	2.22	2.12
TSC	5.68	2.06	GBC	2.20	2.13
MTORA	5.44	2.60	SERCA	2.18	2.18
ULK	4.21	1.92	AMPK	2.17	2.12
IP3	4.06	2.28	CYTCM	2.05	2.03
BID	3.61	1.74	EPAC	1.84	1.84
PKC	3.21	2.53	PLCE	1.80	1.80

**Table 3 ijms-25-11316-t003:** Prior and posterior rate coefficient values and uncertainty ranges for reactions with the most constrained rate coefficients in ascending order (green lowest, yellow highest). The rate coefficients values for the reactions are taken in $\frac{1}{s}$ and $\frac{{\mathrm{cm}}^{3}}{\mathrm{mol}\cdot s}$ units, respectively.

#	Reaction	$\log_{10}k_{\mathrm{ini}}$	$f_{\mathrm{prior}}$	$\log_{10}k_{\mathrm{opt}}$	$f_{\mathrm{posterior}}$	$10^{f_{\mathrm{posterior}}}$
73	ATG5T+BCL2→ATG5_BCL2	6.50	4.00	4.79	0.13	1.35
43	IP3→PIP2	−3.55	4.00	−5.15	0.14	1.37
104	ATG5→REF	−4.55	4.00	−4.51	0.19	1.55
102	REF→ATG5	−15.55	4.00	−14.23	0.20	1.57
109	PKC+CA2IC→PKC_CA2IC	5.44	4.00	4.17	0.26	1.81
10	BCL2_BAX→BCL2+BAX	−3.50	4.00	−5.76	0.26	1.84
63	MTORA→MTOR	−3.55	4.00	−3.45	0.60	3.98
71	AKTA→AKT	−3.53	4.00	−6.30	0.63	4.25
54	EPACA→EPAC	−3.50	4.00	−4.83	0.63	4.26
18	REF→BID	−17.59	4.00	−16.63	0.65	4.49
30	PUMA→REF	−3.92	4.00	−4.99	0.73	5.36
29	BCL2_PUMA→PUMA+BCL2	−3.08	4.00	−3.34	0.87	7.36
69	RHEBA+MTOR→RHEBA+MTORA	6.45	4.00	8.16	0.88	7.65
28	PUMA+BCL2→BCL2_PUMA	7.22	4.00	7.91	0.89	7.70
9	BCL2+BAX→BCL2_BAX	6.54	4.00	3.18	0.94	8.77
8	P53A_BCL2→P53A+BCL2	−6.78	4.00	−6.32	0.97	9.32
44	CA2IC+CAMKKB→CA2IC+CAMKKBA	5.44	4.00	3.12	0.99	9.73
34	CA2IC+SERCA→CA2ER+SERCA	7.01	4.00	3.47	1.00	10.00
45	K+CAMKKBA→AMPKA+CAMKKBA	6.44	4.00	6.05	1.00	10.00

## Data Availability

Data are contained within the article and Appendix A.

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
