# Peer review of "Basal State Calibration of a Chemical Reaction Network Model for Autophagy"

_ijms, 2024, doi:10.3390/ijms252011316_

Round 1
Reviewer 1 Report
Comments and Suggestions for Authors
The study evaluates and calibrates a chemical reaction network model originally developed by Bing Liu et al. to explore autophagy. In their evaluation, initial random concentration conditions were used to predict the behavior of 34 key species. Errors in the model were significant for more than half of species, especially BAX, indicating a need for recalibration. Sensitivity analysis identified 101 influential reactions, and optimization reduced the overall error. While this work reinforces the importance of dynamic modeling in elucidating the mechanisms underlying autophagy, some issues still need to be addressed to enhance the clarity and impact of the work.
1. While the manuscript discusses the importance of dynamic models in understanding autophagy, it could benefit from a clearer justification of why the specific structure of the proposed model was chosen.
2. The initial protein concentrations are critical in reaction network models. Given that initial concentration values had to be defined independently due to the unavailability of the original model's source code, it would be helpful to provide a detailed account of how these values were determined. How do the authors account for the uncertainty introduced by relying on these new literature sources for initial concentrations? Will the variations in these concentrations from different sources affect the model's predictions?
3. The assumption of a cell volume of 10-12 liters seems general. Was this value derived from a specific type of cell, or is it an average across the literature? How sensitive is the model to changes in cell volume?
4. The study assumed a uniform large uncertainty (
4 orders of magnitude) for all reactions. Would refining these assumptions based on biological data for certain well-characterized reactions help improve the model's accuracy?
5. During the optimization of rate coefficients, it was noted that several coefficients were optimized to extreme values, resulting in reactions becoming ineffective. Is there strategies or refinements that might be implemented to avoid getting trapped in local minima during the optimization process?
6. Despite multiple optimizations passes, BAX still has significant errors. Given its importance in apoptosis, what other strategies could the authors use to improve its prediction?
7. Some typos need to be corrected. For example, all the data sources in Table 1 should be provided and properly cited.
Comments on the Quality of English LanguageFurther revisions for grammatical clarity are still needed. And the typos need to be corrected.
Reviewer 2 Report
Comments and Suggestions for Authors
The article presents an interesting research on autophagy and apoptosis through a chemical kinetic model. However, there are some aspects to consider:
1. Structure: The abstract is clear but could be more concise.
2. Methodological details: Although details about the model and its corrections were provided, it would be helpful to include more information on the methodology used for calibration and optimization of the parameters.
3. Discussion of results: You could expand the discussion on the results obtained, especially regarding the clinical impact of the findings.
4. Future implications: Consider adding a section that discusses the potential clinical applications of the model and how it could influence future research.
Overall, the article has the potential to be published, but some revisions and clarifications would increase its quality and impact.
Round 2
Reviewer 1 Report
Comments and Suggestions for Authors
The authors have addressed all my previous comments, and I believe the current paper is suitable for acceptance.
Reviewer 2 Report
Comments and Suggestions for Authors
The manuscript may be accepted for publication in this form.